# Modulating Consciousness through Awareness Training Program and Its Impacts on Psychological Stress and Age-Related Gamma Waves

**DOI:** 10.3390/brainsci14010091

**Published:** 2024-01-17

**Authors:** Kin Cheung (George) Lee, Junling Gao, Hang Kin Leung, Bonnie Wai Yan Wu, Adam Roberts, Thuan-Quoc Thach, Hin Hung Sik

**Affiliations:** 1Centre of Buddhist Studies, The University of Hong Kong, Hong Kong SAR, China; galeng@hku.hk (J.G.); hank.leung@hku.hk (H.K.L.); bonniewu@hku.hk (B.W.Y.W.); hinhung@hku.hk (H.H.S.); 2Singapore-ETH Centre, ETH Zurich, Singapore 138602, Singapore; adamcharles.roberts@sec.ethz.ch; 3Department of Psychiatry, Li Ka Shing Faculty of Medicine, The University of Hong Kong, Hong Kong SAR, China; thach@hku.hk

**Keywords:** aging, electroencephalogram, awareness training program, gamma wave, compensation, meditation

## Abstract

Aging often leads to awareness decline and psychological stress. Meditation, a method of modulating consciousness, may help individuals improve overall awareness and increase emotional resilience toward stress. This study explored the potential influence of the Awareness Training Program (ATP), a form of consciousness modulation, on age-related brain wave changes and psychological stress in middle-aged adults. Eighty-five participants with mild stress were recruited and randomly assigned to ATP (45.00 ± 8.00 yr) or control (46.67 ± 7.80 yr) groups, matched by age and gender. Ten-minute resting-state EEG data, obtained while the participants’ eyes were closed, were collected using a 128-channel EEG system (EGI). A strong positive Pearson correlation was found between fast-wave (beta wave, 12–25 Hz; gamma wave, 25–40 Hz) EEG and age. However, after the 7-week ATP intervention, this correlation became insignificant in the ATP group. Furthermore, there was a significant reduction in stress levels, as measured by the Chinese version of the 10 item Perceived Stress Scale (PSS-10), in the ATP group. These results suggest that ATP may help modulate age-related effects on fast brain waves, as evidenced by the reduced correlation magnitude between age and gamma waves, and lower psychological stress. This suggests that ATP, as a form of consciousness modulation, may improve stress resilience and modulate age-related gamma wave changes.

## 1. Introduction

Aging is a natural part of life, yet it can pose distinctive societal challenges in densely populated countries, such as China (particularly Hong Kong), Japan, Korea, and other developed nations [1]. Due to the considerably longer life expectancies compared to those in the previous century, individuals have more time to experience the challenges of aging, including brain aging. This process leads to slower neuronal processing speed, a subsequent decline in awareness, and emotional disturbances [2]. Stress can also exacerbate aging-related impairment [3,4].

High stress can lead to loss of consciousness, such as syncope, due to an overactive stress response system in the body [5]. Moreover, overstress can negatively impact cerebral function and contribute to cognitive decline associated with aging. Stress-induced neurobiological changes can impede electrical conductivity speed, which is critical for proper brain function [6]. Given that age-related decline in the human brain can begin as early as one’s 30s, early preventative measures may aid in mitigating or slowing the effects of aging on the brain [7]. Stress and aging are linked to changes in brain electrical activity, including decreased gamma wave activity, which may indicate cognitive decline [8].

Stress response is primarily modulated by the interaction between the nervous system and the stress hormone system such as the adrenaline–pituitary axis. Neurological deterioration can manifest in various ways during aging, with the most critical being a slowdown in electrical conduction speed. As brain information processing relies on networks, electroencephalograms (EEGs) have the capacity to capture the dynamics of these networks. Although less prevalent than fMRI in the study of resting state networks, EEGs offer a unique temporal resolution that can elucidate the oscillatory dynamics underlying these networks [9]. Variations in EEG patterns during eyes-closed versus eyes-open resting states imply altering functional connectivity within the default-mode network [10].

Precise neural synchrony is essential for conscious processing, including arousal, perception, attention, and memory, characterized by gamma-range fast oscillations [11,12,13]. A prominent change in the EEGs of older adults is observed in the fast wave band, which includes patterns of narrow-band oscillations in the electrical activity of the human brain with a frequency range of approximately 20–70 Hz [14]. However, research on the relationship between gamma wave activity and aging has yielded contradictory results. Several studies have reported a negative correlation between the power of EEG oscillation in the gamma band and aging, suggesting that the aging process diminishes the brain’s gamma power [15,16]. For instance, Murty et al. [14] conducted a large-scale electroencephalogram (EEG) study comparing gamma oscillations between older and younger adults, finding a decrease in the power of both slow and fast gamma oscillations with age, with more pronounced drops in fast and slow gamma waves. These studies appear to associate aging with reduced gamma power and cognitive performance [15]. However, a few studies have suggested that there is an increase in gamma waves with aging. For example, our previous EEG study found increased gamma waves in older adults compared to the younger control group [17]. Another machine learning study corroborated our finding that aging in middle-aged adults positively correlates with increased power of gamma waves [18].

To explain these conflicting results, some researchers have proposed that the increase in gamma waves as individuals age might be related to heightened psychological stress and alterations in consciousness. Gamma waves may also be linked to increased anxiety, as observed among individuals with anxiety disorders [19]. Furthermore, worry, anxiety, and stress are associated with higher gamma waves in both clinical and non-clinical populations [20,21]. In other words, emotional disturbances in aging brains may result in high levels of gamma waves.

Fortunately, methods such as cognitive training and meditation can help maintain brain functionality during aging and may influence levels of consciousness.

Meditation, a mental practice, has been linked to physiological changes that may help slow the aging process, especially mental aging. Accumulating evidence suggests that meditation might enhance cognitive functions, which are bound to decline with aging, such as attention, memory, and executive functions, as well as alter brain structures susceptible to aging [22,23,24,25,26]. Furthermore, long-term meditators were found to have longer telomeres, the protective caps at the ends of chromosomes that shorten as we age, indicating a potential for reduced cellular aging [27,28]. Notably, since chronic stress accelerates aging processes [29], the stress-reducing effects of meditation, and its potential effects on consciousness, can greatly contribute to its potential anti-aging benefits. Meditation has long been employed as a mental practice for slowing the aging process [2,6,30]. 

Generally, studies have shown that the anti-aging effect of meditation is related to functional changes rather than structural alterations in the brain, resulting in changes in brain network activities and increased meta-awareness and emotional stability [2]. Regarding specific neurological changes, meditation studies typically reveal an enhancement in prefrontal/frontal activity, which is associated with emotional regulation [14]. Meditation is associated with a higher frequency of gamma waves in advanced practitioners [31] and reduced functional activity in the amygdala [22].

However, “meditation” is a broad term encompassing various practices based on different theoretical constructs from previous meditation studies. Given the wide range of meditation methods under the umbrella term “meditation”, at least three potential research issues have arisen. 

First, these different meditative practices stem from unique theoretical and socio-cultural backgrounds and produce distinct neurological and behavioral outcomes. Therefore, specific studies must clarify their unique neural mechanisms to help us to better understand how to apply meditative practices more efficiently. 

Second, because it is unclear whether all types of meditation training can induce an anti-aging effect, a more detailed review of each study is necessary to elucidate the neurological impact of specific meditation methods on brain function.

Third, the most effective form of psychological treatment in clinical psychology is considered culturally congruent [32]. However, there is a lack of culturally congruent meditation research that utilizes a particular population’s cultural and linguistic roots to provide meditation training. For example, Xu and Tracey [33] compared the effectiveness of culturally congruent and Western psychotherapy models in a Chinese sample (*n* = 235) and found that culturally congruent psychotherapy in Chinese culture was significantly more effective than its Western counterparts. Therefore, more research should focus on culturally congruent forms of meditation to reveal their respective neural mechanisms.

Fourth, the scientific research on Buddhist consciousness is limited, yet this concept offers potential benefits for psychological wellbeing. In Buddhist tradition, consciousness is a multifaceted concept with different meanings and interpretations. In Early Buddhist scriptures, consciousness (*viññāna*) is one of the five aggregates that function as a form of bare awareness for registering experiences [34]. The teachings of the Buddha have undergone diversification into numerous lineages as a result of cultural influences and adaptations. Among these, *Mahāyāna* Buddhism has emerged as a profoundly influential sect in various Asian cultures [35]. The term *Mahāyāna*, which translates to “Great Vehicle”, carries the connotation that it sees itself as the all-encompassing path to salvation because *Mahāyāna* emphasizes the concept of bodhisattva, i.e., individuals who are dedicated to helping to liberate others from suffering. *Mahāyāna* Buddhist teaching, especially *Yogācāra* Buddhism, further interprets consciousness as eight consciousnesses, with the first five sense consciousnesses being sensory inputs (sight, smell, touch, hearing, and taste), the sixth consciousness being a “sense-center”, the seventh consciousness being a reflective entity holding a concept of “I”, and the eighth consciousness being a storehouse of past experience and karma [36]. The interpretations of consciousness are designed to help Buddhist practitioners gain insights into the true nature of the mind, thereby using such knowledge for mental cultivation. One way to describe the process of cultivation is as the training or transformation of consciousness to see reality as it is. In the field of Buddhist counseling, there are psychotherapeutic interventions for employing the concept of consciousness as per Buddhism to help patients raise awareness of their cognitive processes in order to foster mental space in making more skillful decisions moment to moment [37,38]. However, such novel applications require more scientific consciousness studies to validate their treatment efficacy.

To address the aforementioned gaps, our study examines the specific neural mechanism of a group intervention based on *Mahāyāna* Chinese Buddhist teaching and meditation practice: the Awareness Training Program (ATP). Unlike other forms of meditation that primarily focus on mindfulness or concentration, this ATP is a group-based Mahayana Buddhist intervention grounded in the Chinese translation of the *Sandhinirmochana Sūtra*, a major *Mahāyāna* Chinese Buddhist text in the *Yogācāra* canon containing core Buddhist teachings on the specific methods of practicing *Samānta* and *Vipassanā* [28]. This approach uniquely combines tranquility and insight techniques, potentially resulting in a broader range of cognitive and emotional benefits. It is important to note that ATP incorporates Buddhist knowledge and practices as a theoretical foundation rather than a religious framework.

Our previous randomized clinical trial (RCT) on ATP demonstrated effectiveness in reducing stress and improving psychological well-being, sense of coherence, and the Buddhist wisdom of non-attachment among 122 working adults [28]. We assume that these benefits may translate into positive changes in brain activity, potentially helping counteract the stress- and anxiety-related processes in aging. Considering this empirical support and previous findings, we hypothesized that (1) participants in the ATP group would show a significant change in gamma power compared to the control group and that (2) the ATP group would have significantly lower stress levels, as measured via the Perceived Stress Scale (PSS), than the control group. We also propose that these changes may improve individuals’ awareness, which may have effects on neural activities and stress levels. This study may contribute new evidence to the literature on mental training and the role of consciousness in stress and related neurophysiological changes.

## 2. Method

In our study, 110 participants (ATP, *n* = 54; waitlist control, *n* = 56) were recruited from the first part of the RCT research [28]; that is, these participants took part in both the previous RCT study on ATP and the current study. The mean ages were 45.00 ± 8.00 years old and 46.67 ± 7.80 years old for the ATP and control groups, respectively. The two groups were matched for age and gender, and all participants had no significant neurological or psychological issues. EEG data were collected from the ATP group after the seven-week ATP course, and the data on the waitlist control group were obtained at a similar time to avoid any temporal bias (Figure 1). This cross-sectional design allowed us to avoid generating a sequence effect for the emotional regulation tasks, as repeated exposure to similar emotional pictures would alter the corresponding brain response. The protocol of the ATP employed is accessible via the doctoral thesis of one of this paper’s authors [2].

### 2.1. EEG Data Collection

The experiment was conducted in a controlled environment in a quiet room specially designed for EEG recordings located in our EEG laboratory. During the experiment, each participant was accompanied by one researcher to ensure the smooth conduct of the session and assist with any issues related to the EEG setup or data recording. EEG data acquisition was carried out using a state-of-the-art 128-channel EGI™ system (Electrical Geodesics, Inc., Eugene, OR, USA). This high-density EEG system allowed for precise measurement of electrical activity across the scalp with high spatial resolution. To ensure optimal signal quality, the impedances of all electrodes were meticulously maintained below 30 KΩ, adhering to the strict requirements of the EGI system. The EEG data were sampled at a high rate of 1000 Hz, which is crucial for capturing the full spectrum of brainwave frequencies. Each EEG recording session, including the resting-state condition, was conducted for a duration of 10 min. Applying this consistent duration across different conditions allowed for a standardized comparison of EEG data. The resting-state condition was particularly important for establishing a baseline measurement of each participant’s brain activity. EEG data were collected while the participants were in a resting state, and the entire recording session was averaged for further analysis. For a more detailed description, refer to our previous ERP study [17].

### 2.2. EEG Data Analysis

The EEG data were processed and analyzed using the EEGLab version 13.0 [31] toolbox based on the MATLAB^TM^ 11.0 platform (MathWorks Inc., Natick, MA, USA). In the preprocessing stage, the EEG data were resampled at 250 Hz, filtered by a finite impulse response filter with a passband of 0.1–100 Hz, and notch-filtered by a short nonlinear infinite impulse response filter with a stopband of 47–53 Hz to reduce any artefacts caused by alternating current. Then, the recorded data’s bad segments (e.g., body/head movement, noticeable muscle artefacts, etc.) were deleted manually, while bad channels were reconstructed with spherical interpolation. Independent component analysis (ICA) was used to remove the components of eye movement, blinking, and other possible forms of artefacts. The data were then reconstructed with the retained components. The ERP data were re-referenced based on left and right mastoids before the statistical analysis.

### 2.3. Psychological Measures

In this study, we used the Chinese version of the 10-item Perceived Stress Scale (PSS-10) [39], which was administered in written form to measure the extent to which life situations are perceived as stressful. This measure was implemented at both pretest and posttest levels. The inventory consists of 10 items ratable on a 5-point Likert scale ranging from 0 (“never”) to 4 (“very often”). Lower scores indicate lower stress levels, and vice versa. In our study, Cronbach’s alpha coefficient was 0.88 at baseline and 0.87 at post-test.

## 3. Results

SPSS version 23.0 was utilized to conduct statistical analyses. Power spectrum analysis and Pearson correlations were used for the EEG data. A mixed between–within subjects ANOVA was conducted to evaluate the effectiveness of the ATP on the participants’ scores on the Perceived Stress Scale (PSS-10) in the intervention and control groups.

We first utilized spectral analysis to evaluate the differences between the ATP and control groups (Figure 2). These data reveal that the ATP group and control group exhibited differences in the amplitude of gamma waves, with the ATP participants being characterized by lower-amplitude gamma waves at the occipital lobe than those of the controls. This distinction between the groups was quite apparent in the participants’ normal resting state.

Our results showed significantly lower gamma wave power, which contrasts with our first hypothesis, which stated that the ATP participants would have a higher gamma wave power than the control group. This finding seems to align with a study by Berkovich-Ohana [40] in which lower frontal gamma in mindfulness meditation practitioners, was observed, potentially attributable to the calming nature of mindfulness practice.

We subsequently calculated Pearson correlations between EEG band power and age for gamma bands for both the ATP and control groups. In the control group, there was a significant positive correlation between age and gamma wave power (r = +0.389, *p* < 0.001). However, this correlation was not significant in the ATP group (r = 0.04, *p* = 0.734) (Figure 3). Moreover, the younger participants displayed higher gamma wave power, and older participants exhibited lower gamma wave power in the ATP group (Figure 4).

To examine our second hypothesis, we performed an analysis of variance (ANOVA) using SPSS 23 to compare the PSS scores between the ATP group and the control group. Prior to the intervention, the average PSS levels for the ATP group were found to be M = 22.15, while the control group exhibited a slightly lower mean score of M = 20.75. Following the intervention, the ATP group displayed a decrease in mean PSS scores to M = 18.15, whereas the control group exhibited a slight increase to M = 19.83. The 54 participants who participated in the ATP (M = 17.96, SD = 5.08), compared to the 56 participants in the control group (M = 19.91, SD = 5.21), demonstrated significantly lower levels of psychological stress as measured via the PSS (F(1, 108) = (3.94], *p* = 0.05). For the treatment effect, the results of the two-way mixed ANOVA indicated a significant interaction between group (intervention and control) and time (pre-test and post-test) with regard to perceived stress (PSS-10: F(1, 113.79) = 19.78, *p* < 0.001, ηp 2 = 0.15). These findings lend support to our second hypothesis.

## 4. Discussion

In this study, we investigated the impacts of age on brain fast wave alterations in middle-aged adults and probed the potential effects of the ATP—a culturally congruent meditation technique—on these age-related neural changes.

One notable finding is the positive correlation between age and gamma waves, which are fast-wave EEG rhythms. Fast waves, including beta and gamma brain rhythms, are typically more susceptible to the effects of aging, as the aging brain often exhibits reduced neural conduction. This finding of increased gamma wave with aging could be compensatory, as an increase in neuron-firing frequency can compensate for delayed neural signal conduction. For instance, the frequency of the beta wave band (12–25 Hz) can approach the range of the gamma band (25–40 Hz). Conventionally, the beta wave oscillatory signal is indicative of focused attention, while the gamma wave signal is linked to intensified cognitive processes and emotional states such as anxiety [41,42]. Indeed, gamma wave spillover may increase anxiety and depression amongst the elderly, reflecting a maladaptive aspect of a compensatory response [43]. However, as our sample primarily consisted of adults, the relationship between increased gamma activity and psychological states in older individuals requires further investigation in older populations.

Delayed neural conduction and the compensatory acceleration of neuron-firing frequency may also result in inconsistency in neural communication and noise in the neural network—factors that could alter one’s state of consciousness. Age-related reduction in neural conduction causes neural signal desynchrony, and higher frequencies of oscillations, such as beta and gamma bands, necessitate low coherence in the cortex. Age-related changes in EEG vary due to coherence cancellation [44]. A rise in neural noise and inconsistency could eventually lead to discrepancies in cognition and emotional regulation among seniors [41], potentially tied to shifts in consciousness.

However, after 7 weeks of ATP training, a significant reduction in perceived stress was observed, and intriguingly, the correlation between gamma wave and age became insignificant in the ATP group. The ATP, as a mental training model, promotes tranquility (*śamatha*) and observation (*vipassanā*). These techniques, which are culturally in alignment with the participants’ Chinese backgrounds and the Chinese Buddhist paradigm, can be seen as capable of modulating one’s state of consciousness. This modulation could have driven the observed changes in gamma waves and stress levels.

We interpret the observed effect as a potential “neutralization” phenomenon, whereby the practice of meditation appears to mitigate or counteract certain adverse effects typically associated with the process of aging. Following this line of thought, one plausible interpretation of this “neutralization” of the effect between age and gamma wave is that ATP might encourage an optimal level of gamma waves in participants. This is because low gamma wave levels are related to cognitive decline, while high gamma wave levels are associated with intensified brain cognitive performance and anxiety. There may be an optimal level of gamma waves that facilitates a certain level of alertness and cognitive acumen. Such a cognitive state, possibly tied to an optimal state of consciousness, does not trigger anxiety responses and may signify the harmonizing effect of ATP on brain activity. The 7-week ATP may have cultivated an optimal state of consciousness that was neither too low (potentially leading to cognitive decline) nor too high (potentially leading to anxiety).

This rationale aligns with the Buddhist idea of the Middle Path (*Majjhimāpaṭipadā*), which is a flexible and stable mental state aiming to avoid extreme thoughts or beliefs, enabling the most skillful decision in a given situation. This state is a balanced and equanimous mindset that is fully aware of happenings without being attached to any particular one. Future studies could delve deeper into this “neutralization” effect and explore the possibility of an optimal or Middle-Path view of gamma waves in aging adults and their ties to consciousness modulation. In this sense, the Buddhist notion of the ‘Middle Path’ aligns with a mental state that avoids extremes, akin to the optimal gamma wave activity that compensates for the effects of aging. This state is marked by an alert and attentive awareness devoid of overactivity that may result in anxiety. ATP may aid in achieving this state by training the mind to sustain a level of alertness for cognitive acuity without overreacting with anxiety or stress.

Interestingly, we found a decrease in gamma waves in the occipital lobe after 7 weeks of the ATP. This finding echoes studies on focused meditation that reported similar results. For example, in a study on mindfulness meditation practitioners, Berkovich-Ohana [40] found lower frontal gamma activity compared to the corresponding control group. On the other hand, like the inconsistent effect of aging on gamma waves, there is controversy regarding the effects of meditation on gamma waves. For example, some studies showed that long-term Buddhist meditators have higher gamma-band activity than control subjects because they can self-induce increased gamma-band oscillations through meditation [8]. This inconsistency in the literature is possibly attributable to the different age ranges of participants, the different brain regions being analyzed, and different types of meditation. Our study suggests that ATP, much like mindfulness meditation, can calm participants, help them cope with stress, and potentially alter their state of awareness/consciousness in a beneficial way.

In particular, previous meditation studies have encompassed various meditation models, such as secular mindfulness techniques [22], Yogic practices [20], Zen meditation [23,45], Indo-Tibetan mind training [46], and *vipassanā* meditation [31]. Meditation may induce vastly different states of mind, and vipassana meditation, in particular, may require significant amounts of mental effort, thereby increasing gamma wave levels [8,47,48]. Mindfulness meditation, however, may promote stability and equanimity of the mind, thus decreasing gamma wave levels [40]. In other words, the ATP requires contemplation and analysis during the meditation process, which provided the grounds for the first hypothesis, while the data showed evidence of the opposite. This finding, namely, ATP participants showing lower gamma wave levels, may suggest that ATP may be more akin to mindfulness meditation, which can calm participants and help them cope with stress. The validation of the second hypothesis also seems to support the participants’ reports of significantly lower perceived stress after treatment. Taken together, these data suggest the effectiveness of ATP in reducing stress and potentially training consciousness from a Buddhist perspective.

## 5. Limitation

It is important to acknowledge the limitations of this study. First, the sample primarily comprised adults, warranting exercising caution when generalizing our findings to older populations, despite the potential benefits of meditation on the aging process. Second, age-related EEG energy changes are more subtle than findings from magnetic resonance imaging (MRI). For example, an age-dependent decrease in coherence can be affected by phase cancellation [44]. Therefore, future studies can examine the effectiveness of ATP using MRI for confirmatory evidence. Third, future research can examine the cognitive performance of control groups and ATP groups of aging adults to shed light on the direct impact of this mental training model. Fourth, the Cronbach’s alpha coefficient level in the current study can be described as being good, but it falls slightly short of an excellent level. Fifth, the sample size may not have been large enough in this study, and future research can aim to investigate the effects of ATP on aging using a larger and more diverse sample. Lastly, our study only captured a cross-sectional angle of ATP participants. Consequently, longitudinal studies of aging adults who practice ATP will likely reveal individual differences and shed light on long-term neurological changes, including potential long-term changes in consciousness.

## 6. Conclusions

In conclusion, we presented a positive correlation between fast brain waves and aging, an effect that becomes insignificant after ATP mental training. This result suggests the potential benefits of ATP as a culturally congruent mental training method for stress reduction and a possible mechanism with which to counter stress in aging. Importantly, our findings also hint at a role of ATP in modulating consciousness, which, in turn, might influence the physiological changes observed in this study. These insights provide valuable information on the specific aspects of ATP that are most beneficial for cognitive aging and stress reduction and could have practical implications for healthy aging, especially during a global pandemic. Individuals working from home, for instance, could benefit from meditation by improving their tolerance of stress and cognitive burden and cultivating long-term healthy aging. Moreover, this group-based training model could be transformed into an online version, thus enhancing accessibility for the elderly during a pandemic. By examining the potential effects of ATP on neural activities, stress levels, and consciousness, our study contributes new evidence to the literature on mental training, aging, and the role of consciousness in emotional and cognitive behaviors.

## Figures and Tables

**Figure 1 brainsci-14-00091-f001:**
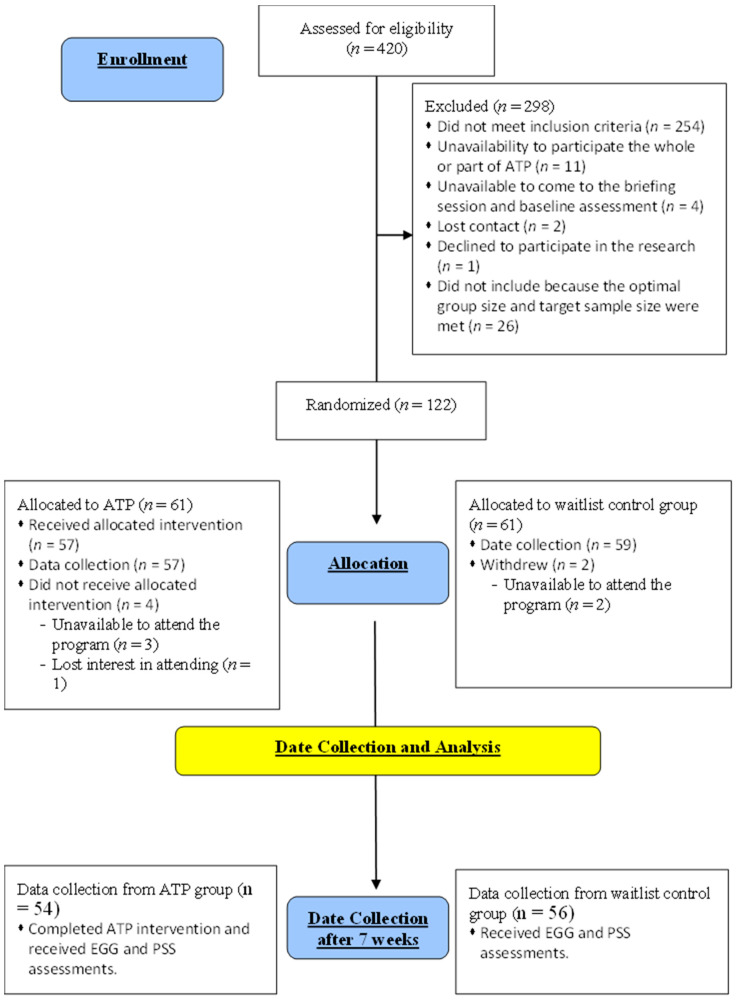
Stages of the ATP study.

**Figure 2 brainsci-14-00091-f002:**
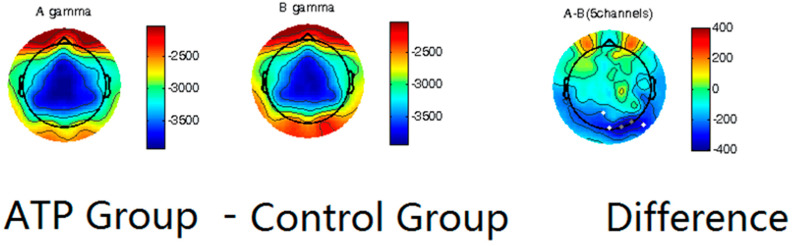
Power spectrum analysis on gamma waves of ATP group (**left** column) and control group (**middle** column), with calculated differences (**right** column). Dots in the last column represent channels with significant differences (*p* < 0.05), and darker dots correspond to smaller *p*-values.

**Figure 3 brainsci-14-00091-f003:**
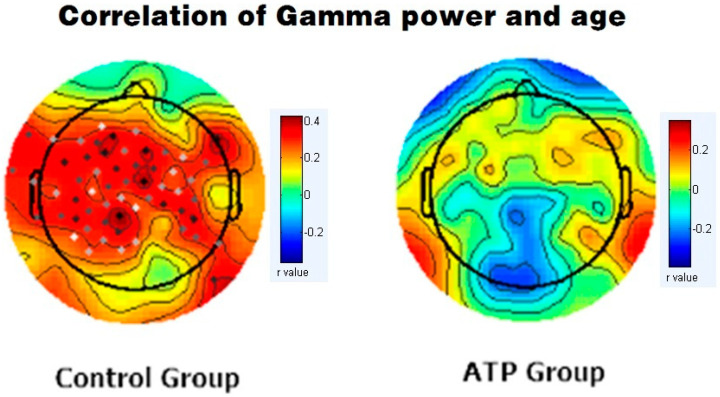
Correlation between gamma band power and age in the control group (**left** column) and the ATP group (**right** column). In the control group, the dots represent significant correlations between gamma band and age (*p* < 0.05), while no significant correlations were identified in the ATP group.

**Figure 4 brainsci-14-00091-f004:**
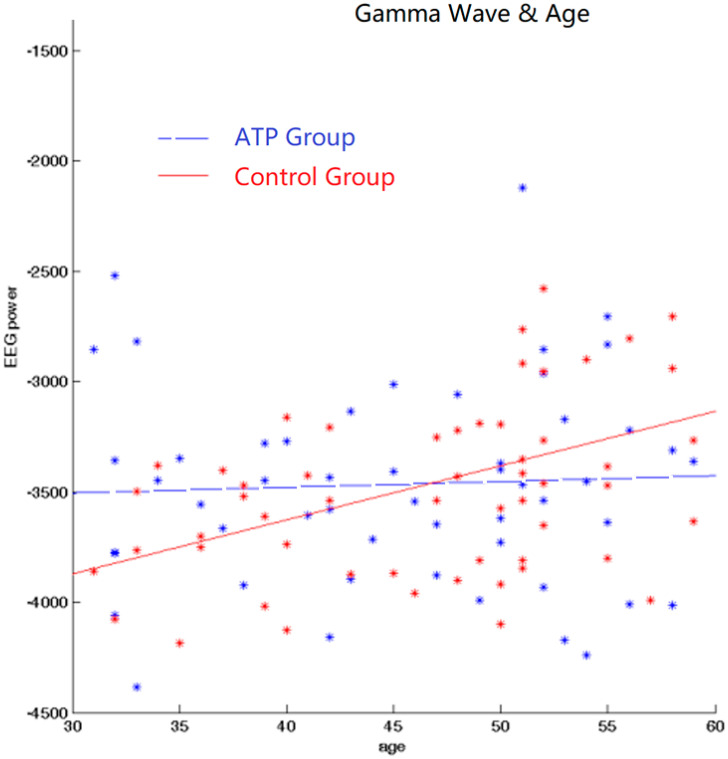
Correlation between gamma and age. There exists a positive correlation between age and gamma wave power for the control group but not for the ATP group. The blue dots and line represent the ATP group; the red dots and line represent the control group.

## Data Availability

The data presented in this study are available on request from the corresponding author. The data are not publicly available due to concerns regarding the confidentiality of the participants.

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
