# Peer review of "Modulating Consciousness through Awareness Training Program and Its Impacts on Psychological Stress and Age-Related Gamma Waves"

_brainsci, 2024, doi:10.3390/brainsci14010091_

Round 1

Reviewer 1 Report

Comments and Suggestions for Authors

Thanks to the authors for sharing their manuscript. I find the research important and interesting. My comments relate to minor revisions:

·         The authors used the 10-item Perceived Stress Scale (short version of the PSS), but in the abstract they write about the Perceived Stress Scale (full version of the PSS). Meanwhile, the full version contains 14 items. Please correct this point in the abstract.

·         Did the respondents fill out the Perceived Stress Scale in English or Chinese? If it is in Chinese, please refer to the Chinese adaptation of the Perceived Stress Scale.

·         The authors describe EEG data analysis, but do not describe data analysis for psychological measures. Further in the Results, they indicate that they used the Pearson coefficient and ANOVA in the SPSS. It seems to me that it is worth describing the data analysis for psychological measures, specifying the version of the SPSS and justifying the choice of statistical methods. Were the data normally distributed, since the authors used parametric methods?

Author Response

Response to Reviewer 1

Thanks to the authors for sharing their manuscript. I find the research important and interesting. My comments relate to minor revisions:

We express our sincere gratitude to Reviewer 1 for providing valuable comments to enhance the quality of our manuscript. In response to each feedback, we provide our detailed response below: 

  • The authors used the 10-item Perceived Stress Scale (short version of the PSS), but in the abstract they write about the Perceived Stress Scale (full version of the PSS). Meanwhile, the full version contains 14 items. Please correct this point in the abstract. 

Thank you for pointing out the inconsistency. Correction made: we used the Chinese version of PSS-10 in the study.

  • Did the respondents fill out the Perceived Stress Scale in English or Chinese? If it is in Chinese, please refer to the Chinese adaptation of the Perceived Stress Scale.

Yes, respondents filled out the Chinese version and we have clarified in the manuscript.

  • The authors describe EEG data analysis, but do not describe data analysis for psychological measures. Further in the Results, they indicate that they used the Pearson coefficient and ANOVA in the SPSS. It seems to me that it is worth describing the data analysis for psychological measures, specifying the version of the SPSS and justifying the choice of statistical methods. Were the data normally distributed, since the authors used parametric methods?

Thank you for the advice. We put in more information to discuss parametric methods used and description of data analysis.

Reviewer 2 Report

Comments and Suggestions for Authors

Modulating Consciousness through Awareness Training and  its impacts on Psychological Stress and Age-Related Gamma  Wave

Authors are right in the introduction about the stress and cerebral activities. However, they need to be more specific about this, adding references , both experimental and clinical.

Line 44- As brain information processing relies on networks, the electroencephalogram (EEG) can readily capture this effect.- Brain networks were studied using fMRI  and several studies have been performed about. Less is known about brain networks using EEG. However, despite this, authors can add more information about brain networks (mainly resting state networks) alterations as studied by EEG.

Line 95 - However, "meditation" is a broad term encompassing various.. I perfectly agree with the authors. However, terms like “ Mahāyāna” needs to be explicated. I am not so sure but “great Vehicle” seems to be the correct translation. It is interesting, but several readers are not familiar with these kind of concepts.

Similarly, I advise to add information about fMRI studies and meditation.

However, the authors highlighted   the effect of meditation on the aging, but adults mainly composed the sample that they recruited. The information about aging should be taken with caution. Indeed, several researchers are interested in the stress and anxiety related processes in the aging.   

In the methods, it is not clear if the present study is a RCT or not. The authors should be clearer about this and I suggest adding a figure to explain the stages of the study.

I also suggest to add info about the ATP protocol. This is crucial for replication of the study with different clinical populations.

PSS-10 – As I read seems that there is no previous validation and adaptation studies of the test. Indeed, the Cronbach alpha is good but not excellent. I suggest to add it in the limitations.

However, did the authors assessed, using PSS, pre and post ATP? Basal stress is relevant and the effect of ATP can be assessed with the above-mentioned test.

Line 224- “To examine our second hypothesis, we calculated the PSS scores for each participant and performed an analysis of variance (ANOVA) using SPSS to compare the PSS score between the ATP group and the control group” Did the authors try to classify the whole sample on the basis of the PSS scores?

Line 241- . Fast waves, such as beta and gamma brain rhythms, are typically more susceptible to the effects of aging- These kind of concept should be clarified. You collected adult participants. Similarly, Line 247 is also not clear.

The concept of neutralization, as suggested and hypothesized by the authors needs to be explicated in a better way.

A specific section about the limitations is also needed.

Author Response

We express our sincere gratitude to Reviewer 2 for providing valuable comments and giving lots of effort to help us improve the quality of our manuscript. In response to each feedback, we provide our detailed response below:

Authors are right in the introduction about the stress and cerebral activities. However, they need to be more specific about this, adding references , both experimental and clinical.

Thank you for pointing it out. We added some descriptions and references to be more specific about stress and cerebral activities.

Line 44- As brain information processing relies on networks, the electroencephalogram (EEG) can readily capture this effect.- Brain networks were studied using fMRI  and several studies have been performed about. Less is known about brain networks using EEG. However, despite this, authors can add more information about brain networks (mainly resting state networks) alterations as studied by EEG.

Well noted! We have addressed it accordingly.

Line 95 - However, "meditation" is a broad term encompassing various.. I perfectly agree with the authors. However, terms like “ Mahāyāna” needs to be explicated. I am not so sure but “great Vehicle” seems to be the correct translation. It is interesting, but several readers are not familiar with these kind of concepts.

Well noted. We have added a very brief explanation of Mahāyāna and key belief in the Line 118. Initially some comments suggested us to not discuss too much details in order to avoid being too “religious.” However, we agree with the point of view of reviewer 2 that it is better to elucidate the content instead of shying away from it.

Similarly, I advise to add information about fMRI studies and meditation.

Yes, we have added more literature review in this part.

However, the authors highlighted   the effect of meditation on the aging, but adults mainly composed the sample that they recruited. The information about aging should be taken with caution. Indeed, several researchers are interested in the stress and anxiety related processes in the aging.  

We agree. We have tried to explicate more on our focus.

In the methods, it is not clear if the present study is a RCT or not. The authors should be clearer about this and I suggest adding a figure to explain the stages of the study.

Yes, thank you for the reminder! We have tried to explain for it in line 160 that we use a cross-sectional design to measure effect of meditation of participants who participated in a RCT. We have just added a figure to make it clear.

I also suggest to add info about the ATP protocol. This is crucial for replication of the study with different clinical populations.

We acknowledge the importance while we do not want to replicate previously published information so we cited a source for our protocol in the manuscript. Hope the reviewer can understand.

PSS-10 – As I read seems that there is no previous validation and adaptation studies of the test. Indeed, the Cronbach alpha is good but not excellent. I suggest to add it in the limitations.

We are using the Chinese version of PSS-10 from the validation study conducted in Hong Kong. Hopefully this translation would be the most culturally relevant to our population as our participants are all from Hong Kong. Here is the article:

Leung, D.Y., Lam, Th. & Chan, S.S. Three versions of Perceived Stress Scale: validation in a sample of Chinese cardiac patients who smoke. BMC Public Health 10, 513 (2010). https://doi.org/10.1186/1471-2458-10-513

We have also added the limitation of Cronbach alpha. Thanks!

However, did the authors assessed, using PSS, pre and post ATP? Basal stress is relevant and the effect of ATP can be assessed with the above-mentioned test.

Yes, we did. We used the baseline data to compare with the 7 weeks follow up on PSS-10. Sorry that we did not make it clear enough.

Line 224- “To examine our second hypothesis, we calculated the PSS scores for each participant and performed an analysis of variance (ANOVA) using SPSS to compare the PSS score between the ATP group and the control group” Did the authors try to classify the whole sample on the basis of the PSS scores?

We are not sure if we understand the question. We matched for age and gender and we compared the PSS-10 scores between the intervention group and control group using follow-up data (7 weeks post-intervention). In this way, we classified participants into intervention group vs. control group and used PSS-10 scores as dependent variable. Would it answer reviewer’s question? As an attempt to further clarify, we rewrote the sentence as “we performed an analysis of variance (ANOVA) using SPSS to compare the PSS score between the ATP group and the control group.”

Line 241- . Fast waves, such as beta and gamma brain rhythms, are typically more susceptible to the effects of aging- These kind of concept should be clarified. You collected adult participants. Similarly, Line 247 is also not clear.

Thank you for pointing out the confusion. We have revised and further discussed in the discussion. Hope it comes across better now.

The concept of neutralization, as suggested and hypothesized by the authors needs to be explicated in a better way.

Thanks for the reminder. Please see if our explication make it clearer.

A specific section about the limitations is also needed.

Yes, we tried to discuss our limitations starting from line 353. We will make it more clear.

Reviewer 3 Report

Comments and Suggestions for Authors

1.      As mentioned in the first line of the Introduction, “Aging is a significant societal problem worldwide”. However, ageing is a natural phenomenon. The sentence can be rephrased.

2.      Very older reference like in line 47 and others can be updated.

3.      There are many religious-based statements. Was it necessary to mention or can it be more focused scientifically?

4.      The sample size is not large enough.

Comments on the Quality of English Language

Moderate editing should be done.

Author Response

We express our sincere gratitude to Reviewer 3 for providing valuable comments to enhance the quality of our manuscript. In response to each feedback, we provide our detailed response below:

  1. As mentioned in the first line of the Introduction, “Aging is a significant societal problem worldwide”. However, ageing is a natural phenomenon. The sentence can be rephrased.

Well noted. Thank you for pointing that out and we have made a correction

  1. Very older reference like in line 47 and others can be updated.

Thank you for the feedback! We have updated our references accordingly, such as adding a paper of Pascal Fries: Rhythms for Cognition: Communication through Coherence.

  1. There are many religious-based statements. Was it necessary to mention or can it be more focused scientifically?

We acknowledge that some scholars perceive a dichotomy between religion and science. It is crucial to clarify that in this study, we incorporate Buddhist knowledge and practices as a theoretical foundation rather than a religious framework. Religion typically entails belief in and reverence for supernatural entities, which are not relevant to the discussion or treatment intervention in our study. Buddhism is widely recognized as a school of thought, and even a school of psychology. The statements we made aimed to provide an explanation of the theoretical foundation in Buddhism, facilitating the rationale behind the intervention. For instance, the exploration of consciousness (viññāna) is a component of Buddhist psychology that does not pertain to supernatural or mystical entities. This issue may be prevalent in various Eastern religions, including Taoism and Confucianism. The philosophical aspects often go unnoticed due to the societal focus on religious practices in many cultures. Nevertheless, academic studies of philosophy have acknowledged Buddhism, Taoism, and Confucianism as legitimate and valuable schools of thought. While Western philosophical schools such as Stoicism and Existentialism have provided the foundational framework for CBT and humanistic/existential therapy, our study endeavors to demonstrate that Buddhist philosophy can serve a similar purpose. We have also added a statement to make clarification.

  1. The sample size is not large enough.

Thank you for your comment. While we did recruit more than 80 participants in the EEG study, not everybody had data for this experiment. We acknowledge the limitation due to the sample size and have now included this point in the Limitations section of our manuscript.

Round 2

Reviewer 2 Report

Comments and Suggestions for Authors

Please correct typos and revise the English language.